# Eflornithine Hydrochloride-Loaded Electrospun Nanofibers as a Potential Face Mask for Hirsutism Application

**DOI:** 10.3390/pharmaceutics15092343

**Published:** 2023-09-19

**Authors:** Shuruq S. Almuwallad, Dunia A. Alzahrani, Walaa S. Aburayan, Ahmed J. Alfahad, Khulud A. Alsulami, Alhassan H. Aodah, Samar A. Alsudir, Sulaiman S. Alhudaithi, Essam A. Tawfik

**Affiliations:** 1Bioengineering Institute, Health Sector, King Abdulaziz City for Science and Technology (KACST), Riyadh 11442, Saudi Arabiasalsadeer@kacst.edu.sa (S.A.A.); 2Advanced Diagnostics and Therapeutics Technologies Institute, Health Sector, King Abdulaziz City for Science and Technology (KACST), Riyadh 11442, Saudi Arabiaaaodah@kacst.edu.sa (A.H.A.); 3Department of Pharmaceutics, College of Pharmacy, King Saud University, Riyadh 11451, Saudi Arabia; salhudaithi@ksu.edu.sa; 4Nanobiotechnology Unit, Department of Pharmaceutics, College of Pharmacy, King Saud University, Riyadh 11451, Saudi Arabia

**Keywords:** electrospinning, nanofibers, face mask application, Eflornithine hydrochloride, drug delivery

## Abstract

Hirsutism is a distressing condition that can affect women’s self-esteem due to the excessive amount of hair growth in different body parts, including the face. A temporary managing option is to develop a self-care routine to remove unwanted hair through shaving or waxing. Laser or electrolysis are alternative methods, but in some cases, the use of medications, such as the topical cream Vaniqa^®^, can help in reducing the growth of unwanted hair. Electrospun fibers have been used in several drug delivery applications, including skin care products, owing to their biocompatibility, biodegradability, high surface area-to-volume ratio, and dry nature that can release the encapsulated drugs with maximum skin penetration. Therefore, polyvinyl pyrrolidone (PVP) fibers were fabricated in combination with hyaluronic acid to deliver the active compound of Vaniqa^®^, i.e., Eflornithine hydrochloride (EFH), as a face mask to inhibit excess facial hair growth. The prepared drug-loaded fibers showed a diameter of 490 ± 140 nm, with an encapsulation efficiency of 88 ± 7% and a drug loading capacity of 92 ± 7 μg/mg. The in vitro drug release of EFH-loaded fibers exhibited an initial burst release of 80% in the first 5 min, followed by a complete release after 360 min, owing to the rapid disintegration of the fibrous mat (2 s). The in vitro cytotoxicity indicated a high safety profile of EFH at all tested concentrations (500–15.625 μg/mL) after 24-h exposure to human dermal fibroblast (HFF-1) cells. Therefore, this drug-loaded nanofibrous system can be considered a potentially medicated face mask for the management of hirsutism, along with the moisturizing effect that it possesses. Topical applications of the developed system showed reduced hair growth in mice to a certain extent.

## 1. Introduction

Hirsutism is a distressing condition characterized by abnormally excessive terminal hair growth in women based on a male pattern distribution, which usually follows androgen distribution [1,2]. The hair is usually thick and dark and distributed throughout the face, neck, chest, tummy, lower back, buttocks, or thighs [3]. Hirsutism affects 5–10% of reproductive-age women worldwide [2]. The etiology of this condition could be polycystic ovary syndrome (PCOS), idiopathic, thyroid or adrenal hormone abnormalities, some medications, or rare tumors [2,4,5], but PCOS is the most common condition [6]. Various approaches have been used for the inhibition and/or treatment of unwanted excessive hair growth, depending on the causes and the severity of the hirsutism. The most common and simple methods are plucking, shaving, waxing, threading, and bleaching. These methods are cheap and easy to be applied, but they have a transient effect and must be repeated frequently [7,8]. All these methods have limitations in terms of effectiveness; yet, different side effects such as scarring, a burning sensation, and irritation can also occur. More effective methods have been used, including electrolysis and laser [9,10], the latter of which is more favorable for patients with light skin and dark hair [11,12].

In addition to the physical methods of hair removal, medications can be used to effectively treat hirsutism. The topical cream Vaniqa^®^, with an active component of Eflornithine hydrochloride (EFH), was approved by the US Food and Drug Administration (FDA) in the 2000s for facial topical application to decelerate facial hair growth [13,14]. EFH works by irreversibly inhibiting the activity of ornithine decarboxylase enzymes in hair follicles, which is essential for hair growth and proliferation [15,16]. This cream is topically applied twice daily at least 8 h apart in a thin layer on the affected areas on the face, where an improvement can be recognized after 6–8 weeks [17,18]. Possible adverse effects of the cream include acne, burning, dry skin, and skin irritation [9,10]. A combination of EFH cream with other hair removal methods such as trimming using an electric clipper or laser treatments has promoted a faster effect and demonstrated a significant improvement [19,20].

Electrospun nanofibers have been used in different biomedical applications, including tissue engineering, wound dressings, biosensors, drug delivery, implants, cosmetics, facial masks, and skin drug delivery [21,22,23,24]. Electrospinning is the process of producing nanofibers through applying a high voltage to a viscous polymer, leading to the evaporation of the solvent and the production of solid fibrous mats [25]. Nanofibers offer many advantages as a drug delivery system, including a high surface area-to-volume ratio, which ensures the efficient delivery of hydrophobic and hydrophilic drugs, and good mechanical properties, biocompatibility, biodegradability, and resemblance to the extracellular matrix (ECM) [24,25,26]. Also, nanofibers can increase patient compliance due to the sustained release of drugs that reduce the frequency of drug application. Nanofibers are also malleable and can be synthesized in different pore sizes [26].

Various natural and synthetic polymers can be electrospun, including polyvinyl alcohol (PVA), poly lactide-co-glycolide (PLGA), poly ε-caprolactone (PCL), hyaluronic acid (HA), and polyvinylpyrrolidone (PVP) [26,27]. PVP is a mucoadhesive and hydrophilic polymer that has been formulated and used as a nanofiber in different biomedical applications and drug delivery systems due to its biocompatibility [27]. PVP nanofibers are biodegradable and provide the immediate and ultrafast release of encapsulated drugs owing to their high porosity, large surface-to-volume ratio, and ultrafast disintegration in water [28]. HA is a versatile natural biopolymer that has been electrospun in combination with other polymers due to its high viscosity and high surface tension in aqueous solutions resulting from long electrostatic interactions and intramolecular hydrogen bonds [29]. Nanofibers prepared by HA have been widely used in a variety of applications, such as in wound healing, as scaffolds, in drug delivery systems, and in cosmetics applications such as facial moisturizers due to their unique properties, such as the resemblance to the ECM and its biodegradability and biocompatibility, hydrophilicity, viscosity, and nonallergic features [29,30,31,32,33]. It has been reported previously that nanofibers prepared from PVA combined with HA exhibit mechanical properties, which were measured with the Young’s modulus, elongation at break (%), and maximum displacement as 70.31 ± 5.53 Mpa, 10 ± 0.10%, and 2.5 mm, respectively [34].

There are different commercially available nanofibrous face mask products, such as the CBDerma-Repair^®^ nanofiber mask (MOIA ELIXIRS^®^, Prague, Czech Republic) that is composed of the cannabidiol drug, vitamin C, vitamin E, HA, and gluconolactone. This dry face mask can be activated while using water, and its 100% biodegradability provides a higher absorption of the active ingredients with maximum skin penetration compared to the standard textile mask [35]. To overcome the limitations of topical cream applications such as discomfort and time consumption, as well as the possible side effects, we encapsulated EFH in nanofibers utilizing the hydrophilic polymer PVP in combination with sodium hyaluronate, i.e., the salt form of HA, to be applied as a face mask for inhibiting unwanted excess hair in hirsutism-affected women.

## 2. Materials and Methods

### 2.1. Materials

Polyvinylpyrrolidone (PVP, average molecular weight ~1,300,000), formic acid (98–100% for LC-MS LiCropur), and ethanol (absolute, ≥99.8%) were obtained from Sigma-Aldrich (St. Louis, MO, USA). EFH monohydrate was purchased from Biosynth Carbosynth (Compton, UK). Sodium hyaluronate (HA) was bought from Lifecore Biomedical (Chaska, MN, USA). Acetonitrile (CHROMASOLV, ≥99.9% for LC-MS) was purchased from Fisher Scientific (Waltham, MA, USA), and distilled water was generated from Milli Q, Millipore (Billerica, MA, USA).

### 2.2. Preparation of Electrospun Fibers

PVP was dissolved at a concentration of 8% (*w*/*v*) in 92% (*v*/*v)* ethanol solution and stirred at room temperature until the formation of a homogenous polymer solution. The formed polymer solution was then mixed with sodium hyaluronate (0.5% *w*/*v*) and EFH (1% *w*/*v*) and stirred for at least 4 h. The electrospun nanofiber was prepared using Spraybase^®^ (Dublin, Ireland) following a procedure developed by Tawfik et al. [36]. In brief, the solution was loaded into a plastic 5 mL syringe attached to a 20G inner-diameter needle. The fibers were produced at a flow rate of 1 mL/h and a constant voltage and tip-to-collector distance of 9.1 kV and 15 cm, respectively. The electrospinning process was run in the dark under controlled conditions: a room temperature of 20–25 °C and relative humidity of 30–45%. Drug-free (i.e., blank) fibers were prepared in similar conditions, except for the addition of EFH.

### 2.3. Surface Morphology of EFH-Loaded Nanofibers

The surface morphology of the electrospun fibers was observed using JEOL scanning electron microscopy (SEM) (JSM-IT500HR, ASIA PTE. Ltd., Singapore). Samples were deposited on SEM stubs using carbon tape and coated with 2-nm-thick platinum utilizing the JEOL auto fine coater (JEC-3000FC, ASIA PTE. Ltd., Singapore). The average diameter of approximately 50 fibers (drug-loaded (DL) and blank) was analyzed using ImageJ software (National Institute of Health, Bethesda, MD, USA).

### 2.4. Solid-State Characterization Using X-ray Diffraction (XRD)

The XRD data were collected using Rigaku miniflex 300/600 (Tokyo, Japan), equipped with a Cu Kα radiation source at an excitation voltage of 40 kV and a current of 15 mA. The samples were fixed on glass holders and scanned at 5°/min between 2θ ranging from 2° to 60°. The XRD patterns were analyzed using OriginPro^®^ 2021 software (OriginLab Corporation, Northampton, MA, USA).

### 2.5. UHPLC-MS/MS Analysis of EFH

The quantification of EFH was achieved using a Shimadzu Nexera-LCMS-8050 instrument. The ultra-high-performance liquid chromatography (UHPLC) system comprised a LC-40D XR binary high-pressure gradient pump with an integrated DGU-405 degassing unit, SIL-40CXR autosampler, and CTO-40C column oven (Shimadzu Scientific Instruments, Kyoto, Japan). UHPLC separation was performed with a Raptor HILIC-Si column (150 × 2.1 mm, 2.7 μm particle size). The oven temperature was set to room temperature, while the mobile phase consisted of LC/MS grade water: 0.1% formic acid (A) and acetonitrile: 0.1% formic acid (B). A linear gradient program was used at a flow rate of 0.4 mL/min: 0.0–3.0 min 95% (B), 3.0–8.0 min from 95% to 5% (B), 8.0–11.0 min from 5% (B), 11.0–12.0 min from 5% to 95% (B), and 12.0–15.0 min 95% (B). EFH quantification was obtained on a triple-quadrupole mass spectrometer that was equipped with an electrospray ionization (ESI) source operated on a collision cell at a gas temperature of 300 °C, sheath gas of 50, aux gas of 10, capillary voltage of 3500 V, and 99.999% argon gas. EFH was detected in the ESI-positive mode at a retention time of *R*_t_ = 6.60 min and quantified using multiple reaction monitoring (MRM). The transition ions (*m*/*z*) of EFH were 183→120 (20 eV) as a quantifier and 183→166 (13 eV) as a qualifier. The acquired data were processed using LabSolutions software, and a standard calibration curve (y = 1E + 07x + 225,721; R^2^ = 0.9998) was created using serial dilution ranges between 5 and 0.04 μg/mL of EFH. The calibration curve was plotted using OriginPro^®^ 2021 software (OriginLab Corporation, Northampton, MA, USA).

### 2.6. Determination of the Encapsulation Efficiency (EE%) and Drug Loading (DL)

Certain weights of EFH-loaded fibers samples were dissolved in 25 mL of distilled water and kept at room temperature for at least 4 h. EFH concentrations were measured using the above-developed UHPLC-MS/MS system, and the EE% and DL were determined by the following equations:(1)$$Encapsulation\;Efficiency\;\%=\frac{Actual\;drug\;amount}{Theoretical\;drug\;amount}\times 100$$
(2)$$Drug\;loading=\frac{Entrapped\;drug\;amount}{Fibers\;Weight}$$

The results represent the mean ± standard deviation (SD) of at least three replicates using OriginPro^®^ 2021 software (OriginLab Corporation, Northampton, MA, USA).

### 2.7. In Vitro Drug Release Study

Certain weights of EFH-loaded fibers were placed into glass containers holding 25 mL of distilled water and shaken at 100 RPM and 37 ± 0.1 °C in a thermostatic shaking incubator (Excella E24 Incubator Shaker Series, New Brunswick Scientific Co., Enfield, CT, USA). Aliquots were withdrawn after 5, 10, 15, 30, 60, 120, 240, and 360 min and replaced with fresh, pre-warmed distilled water at an equivalent volume to maintain the sink conditions. EFH concentrations were measured using the above-developed UHPLC-MS/MS system, and the cumulative release % was determined using the following equation:(3)$$Cumulative\;release\;\%=\frac{Cumulative\;drug\;amount}{Theoretical\;drug\;amount}\times 100$$

The results represent the mean ± SD of at least three replicates. The release curve was plotted using OriginPro^®^ 2021 software (OriginLab Corporation, Northampton, MA, USA).

### 2.8. Disintegration Test of EFH-Loaded Nanofibers

The disintegration of the EFH-loaded and blank fibers was evaluated using a modified method described by Alshaya et al. [37]. A 2 × 2 cm square piece of fibers was placed into a petri dish containing 8 mL of distilled water under gentle shaking (50 RPM) in a thermostatic shaking incubator (Excella E24 Incubator Shaker Series, New Brunswick Scientific Co., Enfield, CT, USA) at 37 °C until the complete detachment of the fibers. The results represent the mean ± SD of at least three replicates using OriginPro^®^ 2021 software (OriginLab Corporation, Northampton, MA, USA).

### 2.9. In Vitro Cytotoxicity Assessment of EFH

The in vitro cytotoxicity of EFH was evaluated using the MTS assay described by Alshaya et al. [37]. HFF-1 cells (ATCC SCRC-1041, human skin fibroblast—Homo sapiens) were subcultured and maintained in Dulbecco’s modified Eagle’s medium (DMEM) supplemented with 10% (*v*/*v*) fetal bovine serum (FBS), penicillin 100 U/mL, and streptomycin 100 µg/mL that were obtained from Sigma-Aldrich (St. Louis, MO, USA). HFF-1 cells were seeded into 96-well plates at a density of 1 × 10^4^ cells per well and incubated overnight at 37 °C and 5% CO_2_. Then, 100 µL of increasing concentrations of EFH that ranged from 15.6 to 500 µg/mL were exposed to the cells for a 24-h incubation period. In addition, 0.1% triton x-100- and DMEM-only treated cells were considered the positive and negative controls, respectively. After incubation for 24 h, the consumed medium was aspirated, and 100 µL of the MTS reagent (CellTiter 96^®^ aqueous one solution cell proliferation assay; Promega, Southampton, UK) was added into each well after mixing with DMEM at a ratio of 1:4 and incubated for at least 2 h at 37 °C and 5% CO_2_. The MTS absorbance was measured at 490 nm using a Cytation 3 absorbance microplate reader (BIOTEK Instruments Inc., Winooski, VT, USA). The cell viability % was determined using the following equation:(4)$$Cell\;viability\;\%=\frac{\left(S-T\right)}{\left(H-T\right)}\times 100$$
where S is the absorbance of the cells exposed to EFH, T is the absorbance of the cells exposed to triton x-100 (positive control), and H is the absorbance of the cells exposed to DMEM (negative control). The results represent the mean ± SD of at least three replicates.

### 2.10. Topical Application of EFH-Loaded Nanofibers for Hair Growth Assessment in Mice

Female C57BL/6 mice, 8–9 weeks old, were anaesthetized using isoflurane, and their hair on the lower dorsal skin was shaved via a clipper (Day 0). Animals were then randomized and allocated into three independent groups (4 animals per group): (i) control (no treatment), (ii) blank (drug-free nanofibers), and (iii) 500 µg of drug-loaded nanofibers. On the following day, daily treatment was initiated and continued for an additional 26 days. Nanofiber treatment was performed by wetting the exposed area of skin via purified water using a sprayer, followed by the application of 1 × 1 cm of rapidly dissolving nanofibers to the skin. Animals were monitored for 15 min immediately after each treatment for any sign of toxicity. To assess the efficacy of the treatment, mice were imaged at various intervals throughout the study using a digital camera. Animals used in this study were bred in-house and maintained under a pathogen-free environment at the College of Pharmacy animal facilities at King Saud University (KSU), and all procedures were performed in the controlled facilities in compliance with the protocol number KSU-SE-22-92 approved by the Research Ethics Committee at KSU.

## 3. Results and Discussions

### 3.1. Surface Morphology of EFH-Loaded Nanofibers

The morphology of the blank (PVP:HA) and drug-loaded (PVP:HA:EFH) fibers were assessed using SEM. Figure 1 shows that both fibrous systems have smooth, un-beaded and nonporous surfaces, confirming their successful fabrication using the electrospinning technique. The mean diameters of the blank and EFH-loaded fibers were calculated as 540 ± 130 nm and 490 ± 140 nm, respectively. Such a similarity in the diameters of both nanofibrous systems was due to the use of similar electrospinning parameters like the voltage, flow rate, and tip-to-collector distance, which is in agreement with our recent study of melittin-loaded PVP nanofibers [38].

### 3.2. Solid-State Characterization Using X-ray Diffraction (XRD)

The XRD patterns of PVP, HA, EFH, their physical mixture (PM), and the blank and drug-loaded nanofibers are presented in Figure 2. The diffractograms of PVP and HA showed a broad halo amorphous pattern, which is consistent with the XRD results of Tawfik et al. [39], Bai et al. [40], Chuc-Gamboa et al. [41], and Chen et al. [42]. The diffraction of EFH displayed several characteristic Bragg reflections, indicating the crystallinity of the drug. Intense Bragg’s reflections for the EFH drug are observed at the 2θ values of 9.77°, 21.2°, and 25.67°, while multiple less intense reflections are shown at 28.04°, 28.07°, 30.13°, 30.94°, 31.79°, 32.05°, 33.41°, 40.57°, and 41.23° (Figure 2). This diffractogram is similar to the EFH XRD results reported by Grewal et al. [43] but with a very slight shifting in some reflections. Distinctive diffraction peaks that represented higher intensities of EFH were also shown in the PM and drug-loaded nanofibers but in low intensities, while the blank fibers had a diffractogram identical to that of PVP and HA, showing a broad halo pattern with no characteristic peaks (Figure 2). The low intense reflections of the drug observed in the EFH-loaded fibers could be an indication of the encapsulation of EFH in the developed nanofibers.

### 3.3. Encapsulation Efficiency (EE%), Drug Loading (DL), and Drug Release of EFH-Loaded Nanofibers

To confirm the encapsulation of EFH and determine the EE%, the amount of encapsulated EFH was analyzed using the UHPLC-MS/MS system. The drug calibration curve was plotted using a serial drug dilution that ranged from 5 to 0.04 μg/mL, which demonstrated good linearity, as shown in Figure 3.

The EE% and DL were calculated to be 88 ± 7% and 92 ± 7 μg/mg, respectively. A prompt release of EFH from PVP/HA electrospun fibers is shown in Figure 4. Approximately 80% of EFH was released within the first 5 min, followed by a complete release of the drug in the following 360 min (Figure 4). This release profile was anticipated, as both PVP and HA are hydrophilic polymers and dissolved very rapidly under the in vitro sink conditions [27,44]. A fast-release profile of the EFH drug from the PVP/HA nanofiber system (80% released within 5 min) is consistent with other studies using hydrophilic PVP nanofibers. This is due to the polymer’s high hydrophilicity, which results in increased polymer–solvent interactions, with rapid solvent absorption causing the matrix to expand to a certain extent, eventually leading to the polymer chain separating from its helical structure. For instance, Li et al. prepared an aloin/polyvinylpyrrolidone (PVP)–aloin/PVP/polylactic acid (PLA)–PLA sandwich nanofiber membrane (APP). Within 10 min, the three-layer nanofibers released 60.49%, followed by a steady release over 9 h [45]. In another study, there was a fast dissolution of the poorly water-soluble drug ibuprofen (IBU) from electrospun hydrophilic polyvinylpyrrolidone (PVP) nanofibers, in which 100.3 ± 4.2% was released in the first minute [40]. Furthermore, core–shell nanofibers combined a filament-forming polymer (PVP K90) with a sweetener (sucralose) on the outside and a core of PVP K10 and the helicid drug on the inside, resulting in rapid drug release within 1 min [46].

### 3.4. Disintegration Test of EFH-Loaded Nanofibers

The disintegration of the blank and EFH-loaded nanofibers was measured to be ≤2 s (Figure 5). These rapid disintegration rates agree with the previous studies of Alshaya et al. [37] and Alkahtani et al. [47], who reported similar quick disintegration profiles of their PVP nanofibers systems.

### 3.5. In Vitro Cytotoxicity Assessment of EFH

EFH cytotoxicity was assessed using the MTS assay, which is one of the most common cell viability assays used to evaluate the toxicity of given compounds. Human dermal fibroblast (HFF-1) cells were exposed to six concentrations of EFH (15.6 to 500 µg/mL) for 24 h to mimic the single daily use of the prepared nanofibrous system. The results showed a very high cell viability% (>90%) at all tested concentrations, which suggests a lack of toxicity of EFH on human dermal fibroblasts, as shown in Figure 6. Since EFH is already FDA-approved as a facial cream [14], this can further prove the safety profile of this drug for topical administration.

### 3.6. In Vivo Hair Growth Assessment of EFH-Loaded Nanofibers

Given the prominent effect of EFH in the management of hirsutism and the potential of electrospun nanofibers as a drug delivery platform, we conducted a pilot study to investigate the impact of topically applied EFH-loaded nanofibers on reducing hair growth in vivo. A clipper was utilized to remove mice hair following a procedure developed by Kumar et al. [7], which is superior to other methods, such as waxing or the use of chemical depilatory cream. The progress of hair growth on the dorsal area of tested mice over time is shown in Figure 7. Hair started to grow on day 11 in some animals of the control group and to a lesser extent in the blank and drug-treated groups. As of day 21, although not significant, treatment with EFH delayed hair growth in some mice. Hair was observed in about 75% of animals in the control group and 66% of mice in the blank-treated group, whereas 50% of the drug-treated animals had noticeable hair.

The slight effect of the developed therapy in this pilot study could be attributed to the fact that EFH does not remove hair; however, it delays its regrowth [16]. Other studies performed histological examinations and hair count tests that indicated the extent of regrowing hair, hair reduction, and hair density of the examined subjects [7,19]. In clinical settings, hair reduction is evaluated using a 4-point Physicians Global Assessment scale: clear or almost clear, marked improvement, improved, and no improvement/worse [16]. Here, the results could be considered as improved, which is an indication of facial hair reduction in visibility. In addition, the dosage regimen used for this study (500 µg, applied once daily) was compared to other topical EFH dosage regimens used in preclinical murine studies in which 50 mg per mouse of EFH 13.9% cream was applied twice a day at least 8 h apart [7]. Therefore, optimizing the dose and dosing regimen may enhance treatment outcomes, which will require further investigation. Moreover, the small number of mice used in this pilot study (3–4 animals per group) might not be adequate to properly reveal the impact of treatment, leaving room for further improvement. The use of EFH 15% cream in clinical studies showed superior results over placeboes in reducing hair growth in women with excessive, unwanted facial hair after 2 to 8 weeks of treatment, based on objective and subjective measures [16]. Furthermore, the nanofiber loaded with EFH was applied to mice’s wettened skin, which dissolved rapidly, as shown in our disintegration test; however, the quantity of percutaneous absorption of the drug into the mouse skin needs to be further tested.

EFH, as an active ingredient in Vaniqa© cream, has been approved by the US FDA and used topically to reduce excessive hair growth on women’s faces [13]. Various clinical trials have combined this drug with other hair removal methods and indicated high hair reductions, including Hamzavi et al., who showed highly improved hair reduction when combining Eflornithine cream with laser treatment [19]. Here, we have loaded PVP nanofibers with the EFH drug and HA as an alternative method for applying the drug to skin. Using the available topical cream could cause skin irritation and dryness, limiting its acceptance by patients [9,10]. Our developed nanofiber system can be applied as a face mask to delay hair growth more comfortably in a prolonged release profile. The skin surfaces of the tested mice did not exhibit signs of local adverse effects, such as inflammation or swelling, throughout the experiment. Further studies are required to determine the mask moisturizing effect and the optimum dose and dose regimen. Compared to other topical drug delivery systems, this nanofiber system allows only active ingredients to be incorporated, with no additives that can irritate the skin.

## 4. Conclusions

Hirsutism is a common and debilitating condition, especially in women with PCOS. There are different managing methods for this condition, including the use of topical cream. Vaniqa^®^ has an active component of EFH shown to reduce excessive hair growth, though with some adverse events, such as skin irritation, a burning sensation, and dryness. Therefore, we aimed to fabricate PVP fibers as a face mask containing HA and EFH drugs to inhibit unwanted hair growth and reduce the skin dryness caused by EFH. EFH-loaded fibers were successfully prepared using electrospinning, with an average diameter of 490 ± 140 nm and smooth surface morphology. The EE% and DL of the drug-loaded fibers were calculated as 88 ± 7% and 92 ± 7 μg/mg, respectively. A burst release of approximately 80% of EFH was released after 5 min, and a complete drug release was demonstrated after 360 min. The obtained release profile of the drug was attributed to the rapid disintegration of the fibrous mat (2 s) and the molecular dispersion of the drug in the nanofiber system, which enhanced the drug solubility. The in vitro cytotoxicity of EFH confirmed the nontoxicity of this drug against human dermal fibroblast cells in a concentration range of 500–15.625 μg/mL. Topical application of the developed system on mice revealed its safety and efficiency in reducing hair growth to some extent. Further work is required to assess the mechanical properties, the moisturizing effect, and the optimum dose and dose regimen of the developed system to be potentially utilized as a nanofiber facial mask.

## 5. Patent

This work was submitted to the Saudi Authority for Intellectual Property, submission number 123447464, dated 10 July 2023.

## Figures and Tables

**Figure 1 pharmaceutics-15-02343-f001:**
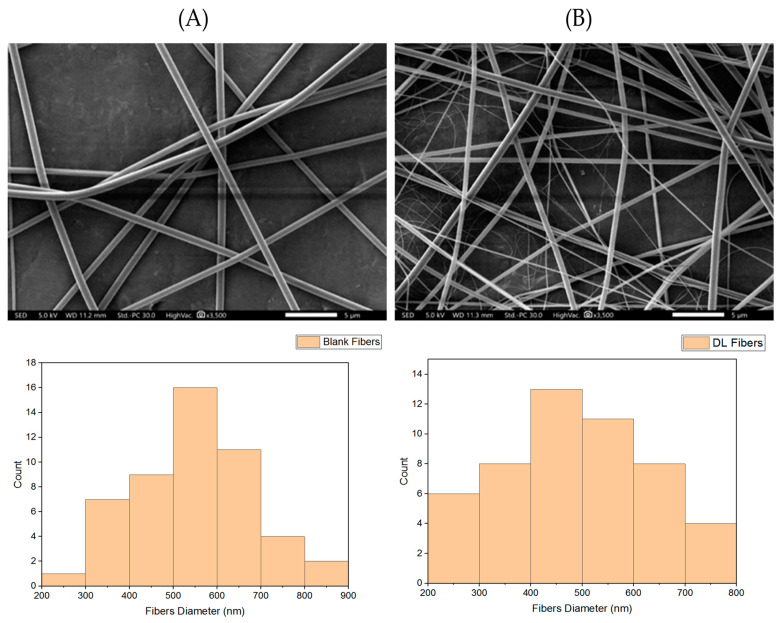
Surface morphology assessment of the blank (**A**) and EFH-loaded (**B**) fibers and their diameter distributions. It shows that both the blank and drug-loaded fibers were successfully prepared with average diameters of 540 ± 130 nm and 490 ± 140 nm, respectively (*n* = 50).

**Figure 2 pharmaceutics-15-02343-f002:**
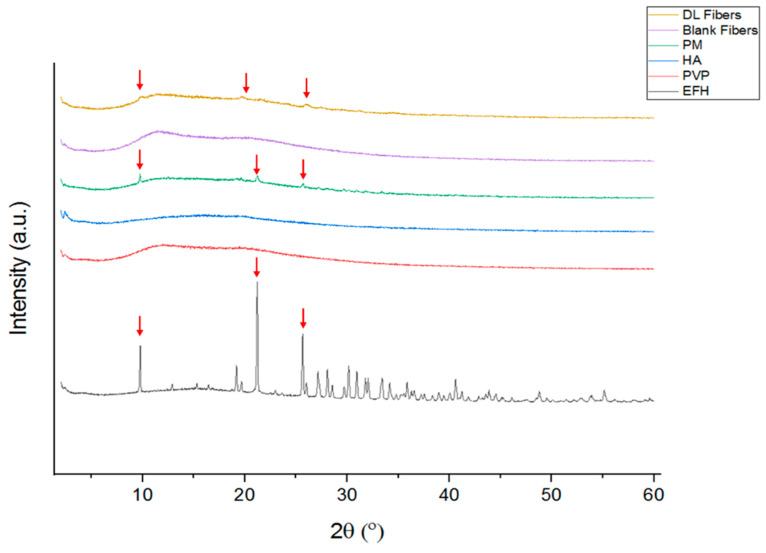
XRD patterns of different polymers, the EFH drug, PM, and electrospun nanofibers. PM is a physical mixture of PVP:HA:EFH, HA is sodium hyaluronate, EFH is Eflornithine hydrochloride, and DL is drug-loaded nanofibers. Red arrow: distinctive peak.

**Figure 3 pharmaceutics-15-02343-f003:**
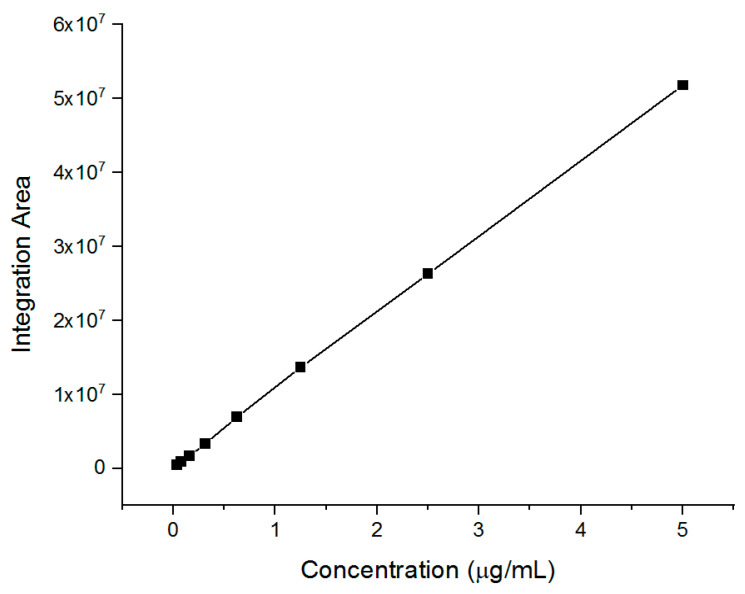
EFH calibration curve using a concentration range of 5–0.04 μg/mL, showing very good linearity (R^2^ = 0.9998) and a regression equation of y = 1 × 10^7^x + 225,721.

**Figure 4 pharmaceutics-15-02343-f004:**
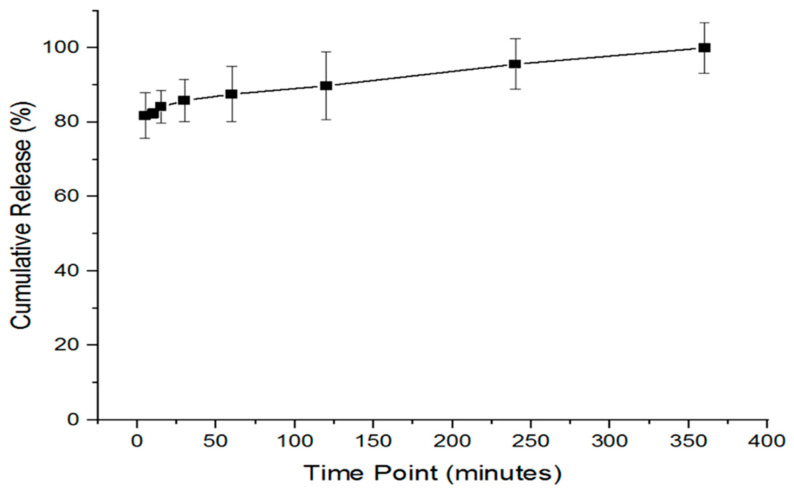
In vitro release profile of EFH-loaded nanofibers, showing that 80% of the drug was released in 5 min and a complete release was observed after 6 h. Results represent the mean ± SD (*n* = 3).

**Figure 5 pharmaceutics-15-02343-f005:**
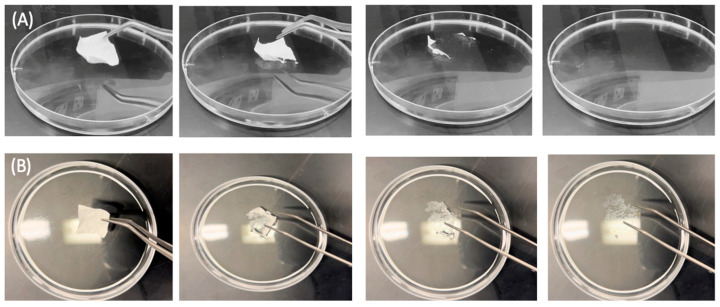
Disintegration test of the blank (**A**) and EFH-loaded (**B**) nanofibers, showing the complete detachment of both fibrous systems after ≤2 s.

**Figure 6 pharmaceutics-15-02343-f006:**
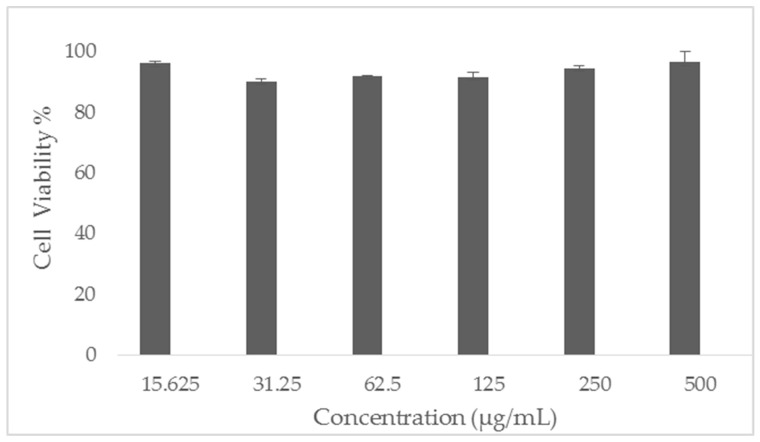
Cell viability % of EFH after 24-h exposure to HFF-1 cells at a concentration range of 15.6 to 500 µg/mL. The data show very high cell viability (>90%) at all tested concentrations. Results represent the mean ± SD (*n* = 3).

**Figure 7 pharmaceutics-15-02343-f007:**
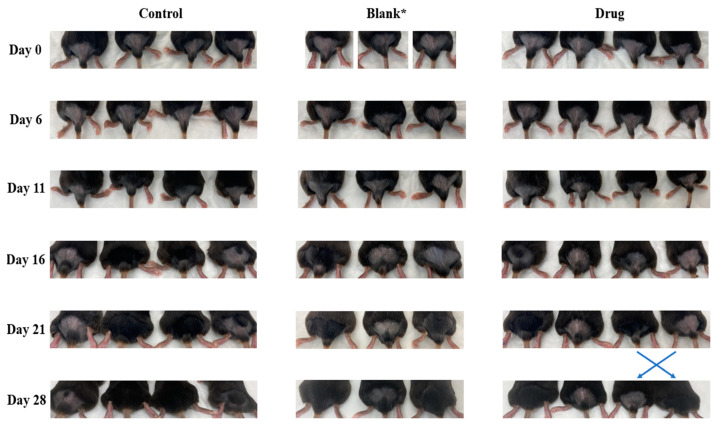
Effect of Eflornithine hydrochloride-loaded nanofibers on hair growth in mice. Images of C57BL/6 mice dorsal skin from various treated groups were acquired by a digital camera on days 0, 6, 11, 16, 21, and 28 (terminal day). * Four animals were originally used in the blank group, but one mouse was deceased on day 2 and thus was removed from the study.

## Data Availability

The authors confirm that the data supporting the findings of this study are available within the article.

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
