# Peer review of "Eflornithine Hydrochloride-Loaded Electrospun Nanofibers as a Potential Face Mask for Hirsutism Application"

_pharmaceutics, 2023, doi:10.3390/pharmaceutics15092343_

Round 1
Reviewer 1 Report
The article is interesting and has some aspects of scientific novelty. After making changes, it can be published. - PVP lacks polydispersity in the Materials and Methods section - some figures lack statistical testing of the results' significance (fig. 6) - the article lacks thermal measurements of the obtained fibres and determination of mechanical properties. Is the strength of the non-woven fabrics sufficient for the proposed application? After making these changes, the article should be reviewed again.
Author Response
We thank the reviewers for their thoughtful and helpful comments. We respond to each point below; their suggestions have undoubtedly improved the paper, for which we are very grateful.
Reviewer #1:
- PVP lacks polydispersity in the Materials and Methods section
Thank you for the suggestion. It is not common in several previous studies on PVP that the authors add the polydispersity of the polymer in the materials section. In addition, there is no information about the polydispersity of PVP on the manufacturer’s website. Measuring the polydispersity using an instrument may vary from one DLS instrument to the other. Therefore, we could not add this parameter in the Materials section.
- Some figures lack statistical testing of the results' significance (fig. 6)
Thank you for this comment. The statistical analyses that were performed in this study focused on calculating the mean, standard deviation (SD), coefficient of determination (R2) and regression equation all using the OriginPro® 2021 software (OriginLab Corporation, Northampton, MA, USA). Comparing the results statistically will not be significant in this study, as we have only one drug-loaded system.
- The article lacks thermal measurements of the obtained fibres and determination of mechanical properties.
Thank you for your comment. The DSC and TGA will not add any significant information especially since we measured the XRD and determined the molecular dispersion, which is a very crucial feature in the electrospinning process. Regarding the mechanical properties, we acknowledge the importance of measuring the nanofibrous systems mechanically, as the suggested application will be on the face after moisturizing it with water. A statement in the conclusion was added, to acknowledge this as future recommendation. In addition, we added in the introduction section a previous study that showed the mechanical strength of another hydrophilic polymer (PVA) with HA as follows: 'It has been reported previously that nanofibers prepared from PVA combined with HA exhibit mechanical properties, which were measured as Young's modulus, elongation at break (%), and maximum displacement as 70.31 ± 5.53 Mpa, 10 ± 0.10 % and 2.5 mm, respectively’.
- Is the strength of the non-woven fabrics sufficient for the proposed application?
In addition to the above response, the disintegration of the fibers was tested, and it showed that the fibers were able to disintegrate very fast (2 seconds), which is another important mechanical-related feature, as the intended future application of these fibers should allow the ultrafast disintegration of the fibers to release the drug and to moisturize the face.
Reviewer 2 Report
In this work, the authors fabricated PVP fiber as a face mask containing HA and EFH drug to inhibit unwanted hair growth and reduce the skin dryness caused by EFH. EFH-loaded fibers were successfully prepared using electrospinning, with an average diameter of 490 ± 140 nm and a smooth surface morphology. The EE% and DL of the drug-loaded fibers were calculated as ~88% and ~92 g/mg, respectively. A burst release of approximately 80% of EFH was released after 5 minutes and a complete drug release was demonstrated after 360 minutes. The obtained release profile of the drug was attributed to the rapid disintegration of the fibrous mat (2 seconds) and the molecular dispersion of the drug in the nanofiber system, which enhances the drug solubility. The in vitro cytotoxicity of EFH confirmed the non-toxicity of this drug against human dermal fibroblasts cells in a concentration range of 500-15.625 μg/ml. Topical application of the developed system on mice revealed its safety and efficiency in reducing hair growth to some extent. Overall, this work is quite interesting for the practical hirsutism treatment application.
1. The manuscript has many extra spaces, which make it hard to read. Also, too many grammar issues in the context.
2. Figure 7. Authors can cut the Photo of Day 28-Drug group, to avoid the misleadingness.
3. Please add the author contributions.
4. Since the PVP and HA are both hydrophilic, will it tend to dissolve (or surface becomes wet) when store in the high-humidity environment?
5. Note that the burst release of EFH (80%) occurs, will that affect the practical hirsutism treatment? I really think the literature comparison should be added in this section.
6. One general question. Compared with the current research about the functional biocompatible hydrogels for hirsutism (such as Bioequivalence of 0.1% mometasone furoate lotion to 0.1% mometasone furoate hydrogel. Australasian Journal of Dermatology, 2016, 57(2), e39-e45. Multiscale bilayer hydrogels enabled by macrophase separation. Matter, 2023, 6(5), 1484-1502.), what is the innovative potential of electrospun nanofibers?
1. The manuscript has many extra spaces, which make it hard to read. Also, too many grammar issues in the context.
Author Response
We thank the reviewers for their thoughtful and helpful comments. We respond to each point below; their suggestions have undoubtedly improved the paper, for which we are very grateful.
Reviewer #2:
- The manuscript has many extra spaces, which make it hard to read. Also, too many grammar issues in the context.
We apologize for this. The layout of the manuscript was changed to the pharmaceutics style and the whole manuscript has been proofread.
- Figure 7. Authors can cut the Photo of Day 28-Drug group, to avoid the misleadingness.
Thank you for this comment and sorry for the confusion. This in vivo study was a preliminary study and there is still room to improve it. We are considering adding more parameters such as PK and PD in our next study. For this study, it was important to set up the experiment and to decide on the end point of the future animal study. So, it is better to report it and consider the 21 days as the endpoint in the future in vivo study.
- Please add the author contributions.
The layout of the manuscript was changed to the pharmaceutics style and the author contribution was added at the end of the manuscript. ‘Conceptualization, Shuruq S. Almuwallad and Essam A. Tawfik; Methodology, Shuruq S. Almuwallad, Walaa S. Aburayan, Ahmed J. Alfahad, Khulud A. Alsulami, Alhassan H. Aodah and Sulaiman S. Alhudaithi; Formal analysis, Dunia A. Alzahrani and Walaa S. Aburayan; Investigation, Shuruq S. Almuwallad, Dunia A. Alzahrani, Walaa S. Aburayan, Ahmed J. Alfahad, Khulud A. Alsulami, Alhassan H. Aodah, Samar A. Alsudir and Sulaiman S. Alhudaithi; Resources, Essam A. Tawfik; Writing – original draft, Dunia A. Alzahrani, Ahmed J. Alfahad, Khulud A. Alsulami, Alhassan H. Aodah, Samar A. Alsudir and Sulaiman S. Alhudaithi; Writing – review & editing, Samar A. Alsudir and Essam A. Tawfik; Visualization, Essam A. Tawfik; Supervision, Essam A. Tawfik; Project administration, Dunia A. Alzahrani.’
- Since the PVP and HA are both hydrophilic, will it tend to dissolve (or surface becomes wet) when store in the high-humidity environment?
Yes, this is true. The intention application of this nanofibrous system will be on the face after moisturizing it with water. The future manufactured finished product should consider packing the product in an airtight sachets-like storage condition. Also, we reported in the introduction that a dry mask commercially available product CBDerma-Repair® nanofiber mask (MOIA ELIXIRS®, Czech Republic) can be activated using water, similar to the intended use of our product. In addition, the disintegration of the fibers was tested, and it showed that the fibers were able to disintegrate very fast (2 seconds), which is another important feature, as it should allow the fibers to release the drug rapidly and to moisturize the face (PVP and HA).
- Note that the burst release of EFH (80%) occurs, will that affect the practical hirsutism treatment? I really think the literature comparison should be added in this section.
Thank you for this comment. We consider this release profile as an advantage, as the intended application of the product is on the face after moisturizing it with water. Having most of the drug released very fast was due to the ultrafast disintegration property (2 seconds) and the molecular dispersion of the drug (confirmed by XRD in this study). Of course, there is still room for improvement in terms of testing the PK of the drug in an animal model and comparing it with the commercially available cream.
The discussion has been amended in the results section, "Encapsulation Efficiency (EE%), Drug Loading (DL), and Drug Release of EFH-Loaded Nanofibers", as follows:
This release profile was anticipated as both PVP and HA are hydrophilic polymers and could dissolve very rapidly under the in vitro sink conditions. A fast-release profile of the EFH drug from a PVP/HA nanofiber system (80% released within 5 minutes) is consistent with other studies using hydrophilic PVP nanofibers. This is due to the polymer's high hydrophilicity, which results in increased polymer-solvent interaction, with rapid solvent absorption causing the matrix to expand to a certain extent, eventually leading to the polymer chain separating from its helical structure. For instance, Li et al. have prepared Aloin/Polyvinylpyrrolidone (PVP)-Aloin/PVP/polylactic acid (PLA)-PLA sandwich nanofiber membrane (APP). Within 10 minutes, the three-layer nanofibers released 60.49%, followed by a steady release over 9 hours [1]. In another study, there was a fast dissolution of the poorly water-soluble drug ibuprofen (IBU) from the electrospun hydrophilic polyvinylpyrrolidone (PVP) nanofibers in which 100.3 ± 4.2% was released in the first minute[2]. Furthermore, Core-shell nanofibers combine a filament-forming polymer (PVP K90) with a sweetener (sucralose) on the outside and a core of PVP K10 and helicid drug on the inside, resulting in rapid drug release within 1 minute[3].
[1] Li, W. et al. Sandwich structure Aloin-PVP/Aloin-PVP-PLA/PLA as a wound dressing to accelerate wound healing. RSC Adv 12, 27300–27308 (2022).
[2] Bai, Y. et al. Testing of fast dissolution of ibuprofen from its electrospun hydrophilic polymer nanocomposites. Polymer Testing 93, 106872 (2021).
[3] Wu, Y.-H. et al. Fast-dissolving sweet sedative nanofiber membranes. Journal of Materials Science 50, 3604–3613 (2015).
- One general question. Compared with the current research about the functional biocompatible hydrogels for hirsutism (such as Bioequivalence of 0.1% mometasone furoate lotion to 0.1% mometasone furoate hydrogel. Australasian Journal of Dermatology, 2016, 57(2), e39-e45. Multiscale bilayer hydrogels enabled by macrophase separation. Matter, 2023, 6(5), 1484-1502.), what is the innovative potential of electrospun nanofibers.
Referring to the suggested study "Bioequivalence of 0.1% mometasone furoate lotion to 0.1% mometasone furoate hydrogel", mometasone furoate hydrogel formula was used instead of the lotion formulation, which provided 38% more hydration for the skin. The topical use of mometasone is indicated for treating itchy, swollen, or irritated skin and is commonly used for conditions like eczema and psoriasis, but not for treating hirsutism (NHS). However, there are certain sensitive patients have been prescribed this lotion along with laser hair removal treatment to reduce erythema and/ or perifollicular edema [1].
The goal of our project was to develop electrospun PVP/HA nanofibers as a delivery platform for EFH drug, which can be applied as a dry face mask on wetted skin, aiming to improve the management of hirsutism and could provide moisturizing benefits. Since PVP nanofibers have a large surface-to-volume ratio, disintegrated in water instantly (2 seconds), and offered immediate and ultrafast release of the encapsulated drugs (80% of drug released in the first 5 minutes). In addition, dry nanofibers, like the nanofibrous system in this study, do not require additives like preservatives or stabilizers that may irritate some customers.
As drug delivery systems, hydrogels made of natural or synthetic materials are also extensively used in biomedical applications due to their inherent properties of water absorption, swelling, nontoxicity, good biocompatibility, and biodegradability [2]. As a standard synthesis procedure, polymerization and crosslinking are required in one or multiple steps [3]. Various physical and chemical techniques like UV photopolymerization or cross-linking chemicals can be used to design polymer chains with multiple sites for drug binding. To make a stable drug-loaded hydrogel, current and different studies have reported different crosslinking methods, including physical, chemical, and enzyme crosslinking. Some chemical crosslinking reaction methods require harsh conditions in which toxic substances may be introduced, reducing biocompatibility. Hydrogel delivery systems must maintain bioactivity during fabrication, transport, and storage, and both the drug and hydrogel must be physically and chemically stable in the process [2-5].
Comparably, drug-loaded nanofiber formulation is easy to formulate and does not require a specific reaction to stabilize the components. In this project, drug-loaded nanofiber formulations were prepared by blending the PVP and HA with the active ingredient (EFH) in a volatile solvent (ethanol) and electrospinning was used that allow solvent evaporation while producing the nanofibrous mat.
References:
[1] Shokeir, H., Samy, N., Mahmoud, H. & Elsaie, M. L. Evaluation of Topical Capislow Extract and Long Pulsed Nd-YAG Laser in the Treatment of Idiopathic Hirsutism. J Lasers Med Sci 9, 128–133 (2018).
[2] Lei, L. et al. Current Understanding of Hydrogel for Drug Release and Tissue Engineering. Gels 8, (2022).
[3] Onaciu, A., Munteanu, R. A., Moldovan, A. I., Moldovan, C. S. & Berindan-Neagoe, I. Hydrogels Based Drug Delivery Synthesis, Characterization and Administration. Pharmaceutics 11, (2019).
[4] Parhi, R. Cross-Linked Hydrogel for Pharmaceutical Applications: A Review. Adv Pharm Bull 7, 515–530 (2017).
[5] Khan, F. et al. Synthesis, classification and properties of hydrogels: their applications in drug delivery and agriculture. J. Mater. Chem. B 10, 170–203 (2022).
Reviewer 3 Report
Recommendation: Major Revision
Comments:
1. Why did the author use HA in this study?
2. Introduction: Drug delivery applications of electrospun nanofibers should be highlighted with the following reference.
- https://doi.org/10.3390/pharmaceutics11070305
3. Experimental: why did the author carry out the electrospinning in dark conditions? Is there any effect of light on the fabrication process?
4. Fig 1A shows the diameter reduction in the composite nanofibers. It should not be explained as a similarity. Although the Process parameters in electrospinning were similar in both cases, the solution parameter may differ. Please check the conductivity and viscosity of the solutions and revise the statement.
5. The mechanical properties of the as-prepared nanofibers membranes should be reported.
Author Response
We thank the reviewers for their thoughtful and helpful comments. We respond to each point below; their suggestions have undoubtedly improved the paper, for which we are very grateful.
Reviewer #3:
- Why did the author use HA in this study?
Thank you for your comment. We highlighted the benefit of HA in the introduction section ‘HA, a natural polymer, has been used as a facial moisturizer due to its hydrophilicity, viscosity and nonallergic features’. We amended this section in the introduction as follows ‘HA is a versatile natural biopolymer that has been electrospun in combination with other polymers due to its high viscosity and high surface tension in aqueous solutions resulting from the long electrostatic interactions and intramolecular hydrogen bonds. Nanofibers prepared by HA have been widely used in a variety of applications, such as in wound healing, as scaffolds, in drug delivery systems, and in cosmetics applications such as facial moisturizers due to their unique properties such as the resemblance to ECM and its biodegradability and biocompatibility, hydrophilicity, viscosity and nonallergic features.
- Introduction: Drug delivery applications of electrospun nanofibers should be highlighted with the following reference. https://doi.org/10.3390/pharmaceutics11070305
The third paragraph in the introduction was amended to include the suggested study as follows: 'Electrospun nanofibers have been used in different biomedical applications including tissue engineering, wound dressings, biosensors, drug delivery, implants, cosmetics, facial masks and in skin drug delivery. Electrospinning is a process of producing nanofiber through applying a high voltage to a viscous polymer leading to the evaporation of solvent and the production of solid fibrous mats. Nanofibers offer many advantages as a drug delivery system including a high surface area-to-volume ratio which ensures efficient delivery of hydrophobic and hydrophilic drugs and good mechanical properties biocompatibility, biodegradability and resemblance to the extracellular matrix (ECM)’
- Experimental: why did the author carry out the electrospinning in dark conditions? Is there any effect of light on the fabrication process?
There is no information about the light sensitivity of the used drug “Eflornithine Hydrochloride monohydrate”. However, the drug was received in a dark amber glass, and the dark condition was considered in case of the sensitivity of the drug to light.
- Fig 1A shows the diameter reduction in the composite nanofibers. It should not be explained as a similarity. Although the Process parameters in electrospinning were similar in both cases, the solution parameter may differ. Please check the conductivity and viscosity of the solutions and revise the statement.
Thank you for this comment. Please note that the diameter of the drug-loaded fibers was slightly reduced than the blank fibers. However, if you noticed the SD for both systems, the difference is not significant. Measuring the conductivity and viscosity of the polymer solution will not be required in this study, as the spraybase electrospinning setup comes with a camera that can easily optimize the cone-jet, and the nanofibrous systems were initially optimized. The SEM images showed the successful preparation criteria of both systems and the voltage was stable at 9.1 KV even after adding the drug at a concentration of 1%.
- The mechanical properties of the as-prepared nanofibers membranes should be reported.
Regarding the mechanical properties, we acknowledge the importance of measuring the nanofibrous system mechanically, as the suggested application will be on the face after moisturizing it with water. A statement in the conclusion was added, to acknowledge this as future recommendation. In addition, we added in the introduction section a previous study that showed the mechanical strength of another hydrophilic polymer (PVA) with HA as follows: 'It has been reported previously that nanofibers prepared from PVA combined with HA exhibit mechanical properties, which were measured as Young's modulus, elongation at break (%), and maximum displacement as 70.31 ± 5.53 Mpa, 10 ± 0.10 % and 2.5 mm, respectively'.
Round 2
Reviewer 1 Report
The article is ready for publication.
Reviewer 3 Report
The paper can be accepted.